 SHORT REPORT

# The pelvic organs receive no parasympathetic innervation

Margaux Sivori[1†], Bowen Dempsey[2†], Zoubida Chettouh[1], Franck Boismoreau[1], Maïlys Ayerdi[1], Annaliese Eymael[2], Sylvain Baulande[3], Sonia Lameiras[3], Fanny Coulpier[4,5], Olivier Delattre[6], Hermann Rohrer[7], Olivier Mirabeau[1‡§], Jean-François Brunet[1*‡]

[1]Institut de Biologie de l'ENS (IBENS), Inserm, CNRS, École normale supérieure, PSL Research University, Paris, France; [2]Faculty of Medicine, Health & Human Sciences, Macquarie University, Macquarie Park, Sydney, Australia; [3]Institut Curie, PSL University, ICGex Next-Generation Sequencing Platform, Paris, France; [4]GenomiqueENS, Institut de Biologie de l'ENS (IBENS), Département de biologie, École normale supérieure, CNRS, INSERM, Université PSL, Paris, France; [5]Inserm U955, Mondor Institute for Biomedical Research (IMRB), Creteil, France; [6]Institut Curie, Inserm U830, PSL Research University, Diversity and Plasticity of Childhood Tumors Lab, Paris, France; [7]Institute of Clinical Neuroanatomy, Dr. Senckenberg Anatomy, Neuroscience Center, Goethe University, Frankfurt am Main, Germany

*For correspondence:
jfbrunet@biologie.ens.fr

†These authors contributed equally to this work
‡These authors also contributed equally to this work

Present address: §Institut Pasteur, Université Paris Cité, Bioinformatics and Biostatistics Hub, Paris, France

Competing interest: The authors declare that no competing interests exist.

**Abstract** The pelvic organs (bladder, rectum, and sex organs) have been represented for a century as receiving autonomic innervation from two pathways – lumbar sympathetic and sacral parasympathetic – by way of a shared relay, the pelvic ganglion, conceived as an assemblage of sympathetic and parasympathetic neurons. Using single-cell RNA sequencing, we find that the mouse pelvic ganglion is made of four classes of neurons, distinct from both sympathetic and parasympathetic ones, albeit with a kinship to the former, but not the latter, through a complex genetic signature. We also show that spinal lumbar preganglionic neurons synapse in the pelvic ganglion onto equal numbers of noradrenergic and cholinergic cells, both of which therefore serve as sympathetic relays. Thus, the pelvic viscera receive no innervation from parasympathetic or typical sympathetic neurons, but instead from a divergent tail end of the sympathetic chains, in charge of its idiosyncratic functions.

## eLife assessment

This **useful** study compares gene expression patterns among different autonomic ganglia and will be of interest to developmental neuroscientists and neurophysiologists. The study expands the database of genes expressed by subpopulations of autonomic neurons in ganglia, a key step in decoding their developmental origins and physiological functions. The evidence supporting the alternative view that the pelvic ganglionic neurons are actually modified sympathetic neurons is **incomplete** and may cause confusion, given the enrichment of cholinergic neurons, as well as the large number of molecular and functional differences known to be present between cranial and sacral neurons.

## Introduction

The pelvic ganglion is a collection of autonomic neurons, close to the walls of the bladder, organized as a loose ganglionated plexus (as in humans) or a bona fide ganglion (as in mouse). It receives input from preganglionic neurons of the thoracolumbar intermediate lateral motor column, a sympathetic

pathway that travels through the hypogastric nerve. A second input is from the so-called 'sacral parasympathetic nucleus' that projects through the pelvic nerve. The assignment of these two inputs to different divisions of the autonomic nervous system by *Langley, 1921*; *Langley, 1899*, largely based on physiological observation on external genitals which were never generally accepted (reviewed in *Espinosa-Medina et al., 2018*), has led, in turn, to propose that the pelvic ganglion, targeted by both pathways, is of a mixed sympathetic/parasympathetic nature (*Kuntz and Moseley, 1936*). Of note, this unique case of anatomic promiscuity between the two types of neurons poses a challenge for the schematic representation of the autonomic nervous system, so that the pelvic ganglion is most often omitted from such representations (*Espinosa-Medina et al., 2018*). A duality of the pelvic ganglion was also suggested by the coexistence of noradrenergic and cholinergic neurons, which elsewhere in the autonomic nervous system form, respectively, the vast majority of sympathetic ganglionic cells and the totality of parasympathetic ones. Here, we directly explore the composition of the mouse pelvic ganglion in cell types, using single-cell transcriptomics, and compare it to sympathetic and parasympathetic ganglia.

## Results

We isolated cells from several autonomic ganglia of postnatal day 5 mice: the stellate ganglion and the lumbar chain (both belonging to the paravertebral sympathetic chain), the coeliac-superior mesenteric complex (belonging to the prevertebral sympathetic ganglia, later refered to as "coeliac"), the sphenopalatine ganglion (parasympathetic) and the pelvic ganglion, and processed them for single-cell RNA sequencing (cf Materials and methods). Neuronal cells segregated in three large ensembles (*Figure 1A*; *Figure 1—figure supplement 1A and B*): one that contained all sympathetic neurons, one that contained all parasympathetic neurons, and the third that contained most pelvic neurons – except one subset that segregated close to the sympathetic cluster. Thus, no pelvic neuron segregates with parasympathetic neurons, but the great majority of them do not segregate with sympathetic neurons either.

The separation, on the Uniform Manifold Approximation and Projection (UMAP), of most pelvic from all other ganglionic cells contrasts with the suite of five developmental transcription factors that we previously reported as differentially expressed between the sympathetic and pelvic ganglia on one hand, and parasympathetic ganglia on the other (*Espinosa-Medina et al., 2016*). To explore this conundrum, we searched for more genes that would help place the pelvic ganglion relative to the sympatho-parasympathetic dichotomy: additional genes that would put the pelvic ganglion in the sympathetic category (as per our previous findings), genes that would put it in a class by itself (as the UMAP suggests), or genes that would split pelvic neurons into parasympathetic-like and sympathetic-like clusters (as the current dogma implies).

In an unbiased approach, we sought genes expressed in a higher proportion of cells in any set of ganglia compared to its complementary set (see 'Materials and methods'). To do justice to the classical notion that the pelvic ganglion is heterogeneous, i.e., mixed sympatho/parasympathetic, we treated the clusters of pelvic ganglionic cells, four in our conservative estimate (P1–4) (*Figure 1A*; *Figure 1—figure supplement 1B*) as four ganglia, alongside the four other ganglia (sphenopalatine, stellate, coeliac, and lumbar) – i.e., we made $(2^8-2)=254$ comparisons and analyzed the 100 'top' genes, i.e., with the highest discrimination score, irrespective of the comparison scored (*Supplementary files 1–3*).

The vast majority of the top 100 genes fell into 7 of the 254 possible dichotomized expression patterns, visualized on a heatmap (patterns I–VI, *Figure 1B and C*; pattern VII, *Figure 1—figure supplement 2*) where cells are grouped by ganglion, and genes by expression pattern (i.e. disregarding their score). These seven patterns can be consolidated into three, among which only the first is informative about a sympathetic or parasympathetic identity of pelvic neurons:

1. Patterns I–IV comprise 39 genes with an opposite status in sympathetic and parasympathetic cells, among which 20 (I+III) are sympathetic-specific and 19 (II+IV) parasympathetic-specific (*Figure 1B and C*). Those which are also expressed in pelvic clusters (I and IV) argue for their sympathetic (respectively parasympathetic) identity, and against the opposite identity; those which are not expressed in pelvic clusters (II and III) argue solely against such an identity. Overall, 32 genes (I+II) argue against a parasympathetic identity of all pelvic clusters, among which 16 genes (I) argue for their sympathetic identity; 7 genes (III +IV) argue against a sympathetic

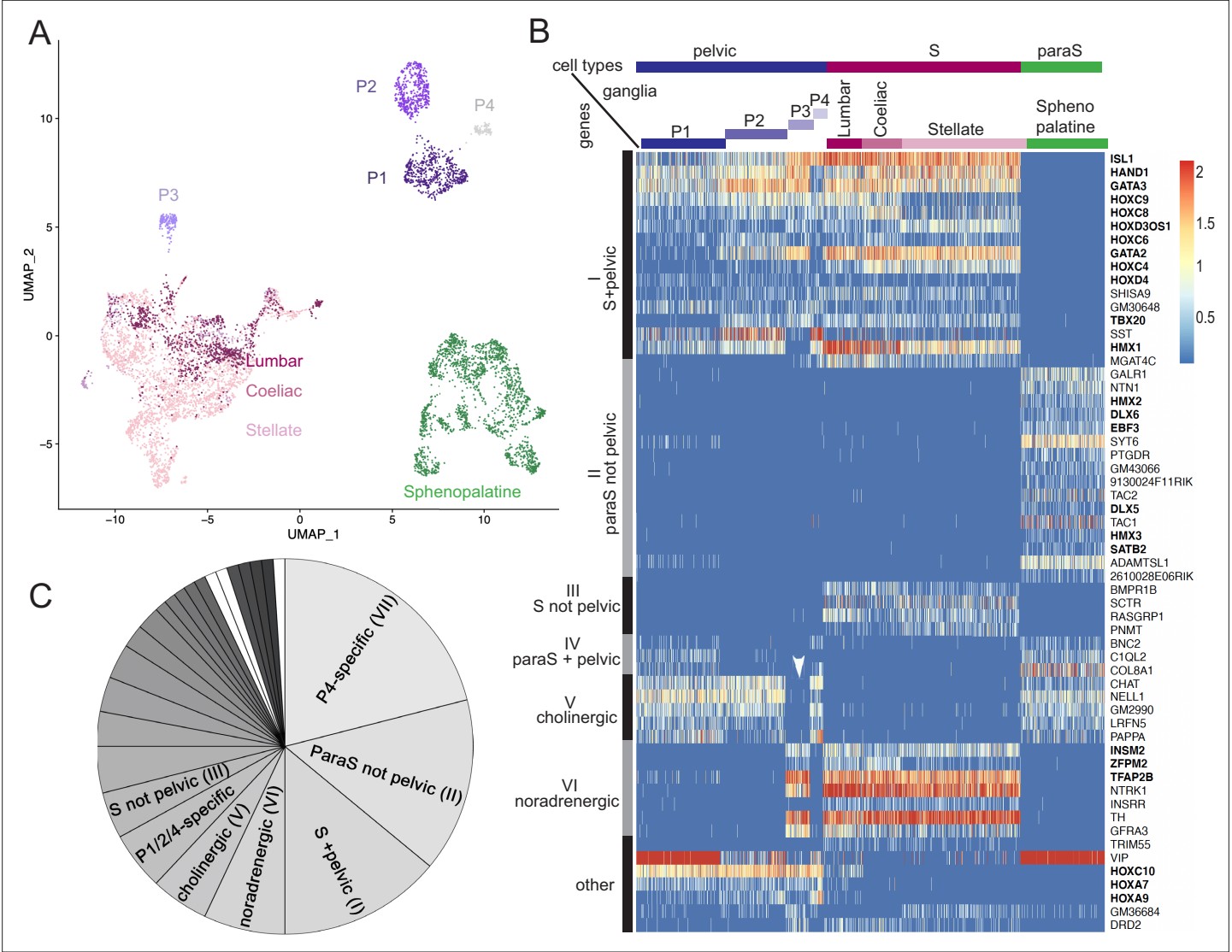

**Figure 1.** The pelvic ganglion does not contain parasympathetic neurons and is made of sympathetic-like neurons. (**A**) Uniform Manifold Approximation and Projection (UMAP) of cells isolated from three sympathetic ganglia (lumbar, stellate, and celiac), a parasympathetic ganglion (sphenopalatine), and the pelvic ganglion dissected from postnatal day 5 mice. The pelvic ganglion is sharply divided into four clusters (P1–4), none of which co-segregates with sympathetic or parasympathetic neurons. (**B**) Heatmap of the highest scoring 100 genes in an all-versus-all comparison of their dichotomized expression pattern among the four ganglia and four pelvic clusters (see 'Materials and methods'), excluding genes specific to the pelvic ganglion (shown in *Figure 1—figure supplement 2*), and keeping only the top-scoring comparison for genes that appear twice. For overall legibility of the figure, the three largest cell groups (lumbar, stellate, and sphenopalatine) are subsampled and genes are ordered by expression pattern (designated on the left), rather than score. 'Cholinergic' and 'noradrenergic' genes are those that are coregulated with *ChAT* or *Th*, regardless of known function. 'Other' refers to various groupings that split sympathetic ganglia and are thus not informative about a sympathetic or parasympathetic identity. Transcription factors are indicated in bold face. White arrowhead: pelvic P3 cluster; S, sympathetic; ParaS, parasympathetic. (**C**) Pie chart of the top 100 genes, counted by expression pattern in the all-versus-all comparison. Genes specific for the P4 cluster dominate (see heatmap in *Figure 1—figure supplement 2*), followed by those which are 'parasympathetic-not-pelvic' and 'sympathetic-and-pelvic' (seen in B). The three genes marked in white (which form group IV: *Bnc2*, *C1ql2*, *Col8a1*) are the only ones that are compatible with the current dogma of a mixed sympathetic/parasympathetic pelvic ganglion, by being expressed in the sphenopalatine ganglion and a subset of pelvic clusters (other than the full complement of cholinergic ones, which define group V).

The online version of this article includes the following figure supplement(s) for figure 1:

**Figure supplement 1.** Uniform Manifold Approximation and Projection (UMAP) of all ganglionic neurons.

**Figure supplement 2.** Genes specific for pelvic ganglionic cells among the top 100 genes of an all-versus-all comparison.

**Figure supplement 3.** Five top genes for each of the four individual pelvic clusters in an all-versus-all comparison.

**Figure supplement 4.** Expression of all *Hox* genes captured by the single-cell RNA sequencing dataset.

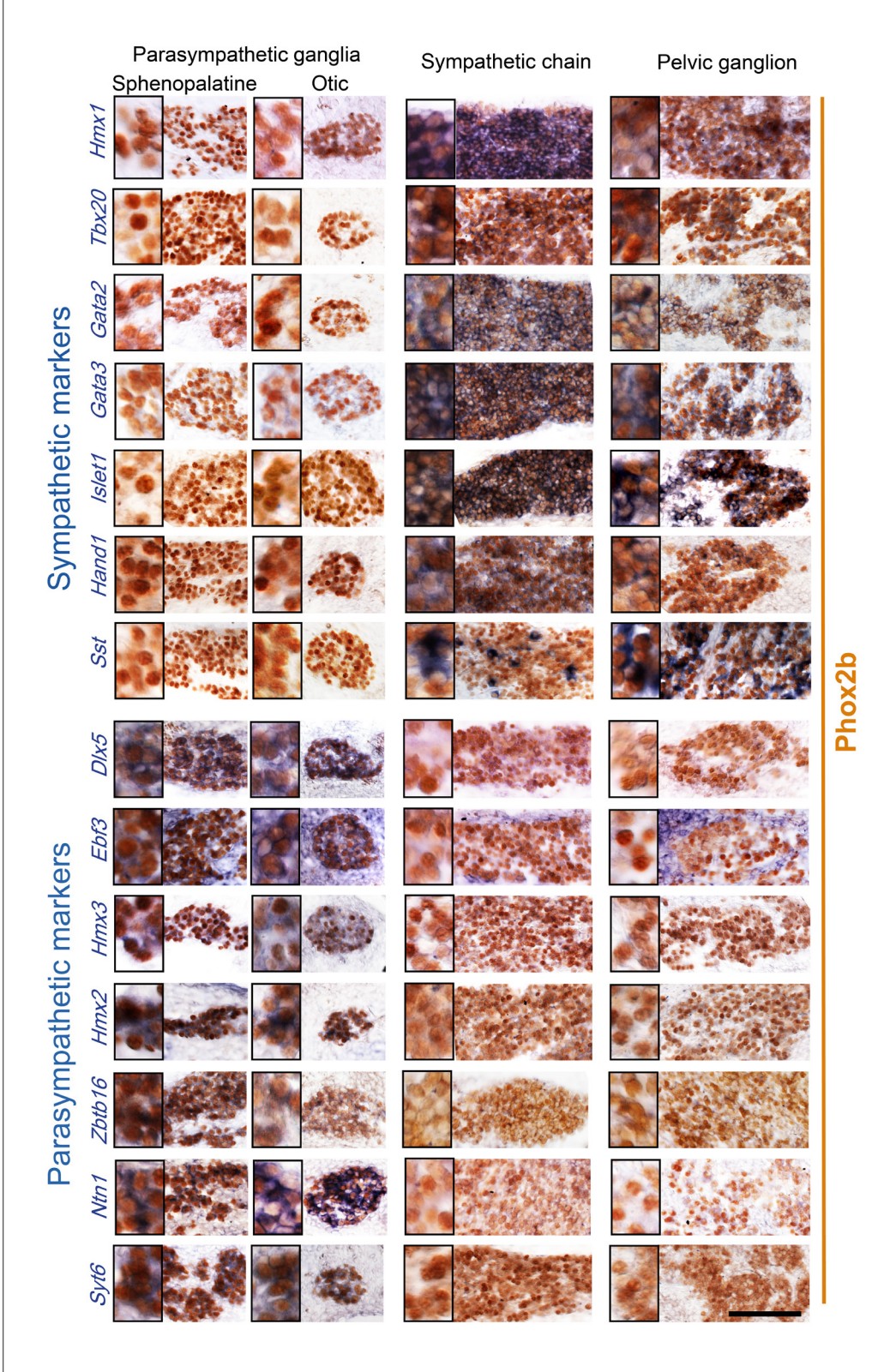

**Figure 2.** Pelvic ganglion cells express sympathetic but not parasympathetic markers. Combined immunohistochemistry for Phox2b and in situ hybridization for seven sympathetic markers (upper panels) including six transcription factors or seven parasympathetic markers (lower panels) including five transcription factors, in two parasympathetic ganglia (sphenopalatine and otic), the lumbar sympathetic chain, and the pelvic ganglion, at

*Figure 2 continued on next page*

*Figure 2 continued*

low and high magnifications (inset on the left) in E16.5 embryos. *Ebf3* is expressed in both, the parasympathetic ganglia and the mesenchyme surrounding all ganglia. *Sst* is expressed in a salt and pepper fashion. *Zbtb16*, a zinc-finger transcriptional repressor, appeared after the 100 highest scorer gene of our screen, but was spotted as expressed in the sphenopalatine in Genepaint. Some transcription factors detected by the RNA sequencing screen at P5 (*Satb2*, *Dlx6*) were expressed below the detection limit by in situ hybridization at E16.5. Scale bar: 100μm.

identity of all clusters, among which 3 genes (IV) argue for a parasympathetic identity of P1 or P4 (which are otherwise not parasympathetic by the criterion of 31 and 32 genes, respectively). We verified expression of 7 sympathetic and 7 parasympathetic markers by in situ hybridization on the pelvic ganglion at E16.5 (*Figure 2*).

2. Patterns V+VI comprise 12 genes that correlate with neurotransmitter phenotype (cholinergic or noradrenergic) by being expressed in the sphenopalatine ganglion and P1, P2, and P4 (5 genes (V), including *ChAT*) or, conversely, in all sympathetic ganglia and P3 (7 genes (VI), including *Th*) (*Figure 1B and C*). These genes point to noradrenergic and cholinergic 'synexpression groups' (*Niehrs and Pollet, 1999*) broader than the defining biosynthetic enzymes and transporters. Neurotransmitter phenotype has been suggested to largely coincide with origin of input, lumbar, or sacral (*Keast, 1995*) which currently defines, respectively, sympathetic versus parasympathetic identity (*Keast, 2006*). However, we find that the lumbar sympathetic pathway targets both cholinergic and noradrenergic pelvic ganglionic cells (in a proportion that reflects their relative abundance), by anterograde tracing (*Figure 3*). Thus, noradrenergic and cholinergic genes are excluded from a debate on the sympathetic or parasympathetic identity of pelvic neurons.

3. Pattern VII involves 42 genes that place all or some pelvic clusters in a class by themselves (*Figure 1—figure supplement 2*), thus neither sympathetic nor parasympathetic (as does the whole transcriptome, as evidenced by the UMAP (*Figure 1A*)), by being expressed, or not expressed, exclusively in them. The cholinergic cluster P4 was particularly rich in genes with such idiosyncratic expression states (20 genes). *Figure 1—figure supplement 3* provides the five top genes of each pelvic cluster.

In the aggregate, the pelvic ganglion is best described as a divergent sympathetic ganglion, devoid of parasympathetic neurons. The fact that few of the top 100 genes (4 genes, III) marked sympathetic neurons to the exclusion of pelvic ones (*Figure 1B*), while many (40 genes, VII) did the opposite (*Figure 1—figure supplement 2*), argues that the pelvic identity is an evolutionary elaboration of a more generic, presumably more ancient, sympathetic one.

A third of the top 100 genes (32 genes) are transcription factors: 6 define a parasympathetic/non{sympathetic+pelvic} identity (*Hmx2*, *Hmx3*, *Dlx5*, *Dlx6*, *Ebf3*, *Satb2*); 12 define a {pelvic+sympathetic}/non-parasympathetic identity (*Isl1*, *Gata2*, *Gata3*, *Hand1*, *Hmx1*, *Tbx20* plus 6 *Hox* genes); 3 correlate with the noradrenergic phenotype (*Tfap2b*, *Insm2*, and *Zfpm2*); 8 (including 6 *Hox* genes) are specific to the pelvic ganglion or some of its clusters (*Figure 1—figure supplement 2*) and 3 *Hox* genes are specific to the pelvic+lumbar ganglia (*Figure 1B*). Nothing is known of the function of the 6 parasympathetic transcription factors, but most of the non-*Hox* {sympathetic+pelvic} ones are implicated in sympathetic differentiation: *Islet1* (*Huber et al., 2013*), *Hand1* (*Doxakis et al., 2008*), *Gata2* (*Tsarovina et al., 2004*), *Gata3* (*Lim et al., 2000*), and *Hmx1* (*Furlan et al., 2013*).

Expression of a shared core of 12 transcription factors by pelvic and sympathetic ganglia, yet of divergent transcriptomes overall, logically calls for an additional layer of transcriptional control to modify the output of the 12 sympathetic transcription factors. Obvious candidates are the 6 *Hox* genes restricted to the pelvic ganglion (*Hoxd9*, *Hoxa10*, *Hoxa5*, *Hoxd10*, *Hoxc11*, and *Hoxb3*) (*Figure 1—figure supplement 2*) and 6 more beyond the 100 top genes, mostly *Hoxb* paralogues (*Figure 1—figure supplement 4*). Given that the parasympathetic and sympathetic ganglia are deployed according to a rostro-caudal pattern (parasympathetic ganglia in register with cranial nerves, sympathetic ganglia with thoraco-lumbar nerves, and the pelvic ganglion with lumbar and sacral nerves; *Figure 4*), the entire taxonomy of autonomic ganglia could be a developmental readout of *Hox* genes (whose multiplicity makes this conjecture hard to test). A role for *Hox* genes in determining types of autonomic neurons would be reminiscent of their specification of other neuronal subtypes in *Drosophila* (*Suska et al., 2011*), *Caenorhabditis elegans* (*Zheng et al., 2015*), and *Mus* (*Coughlan et al., 2019*).

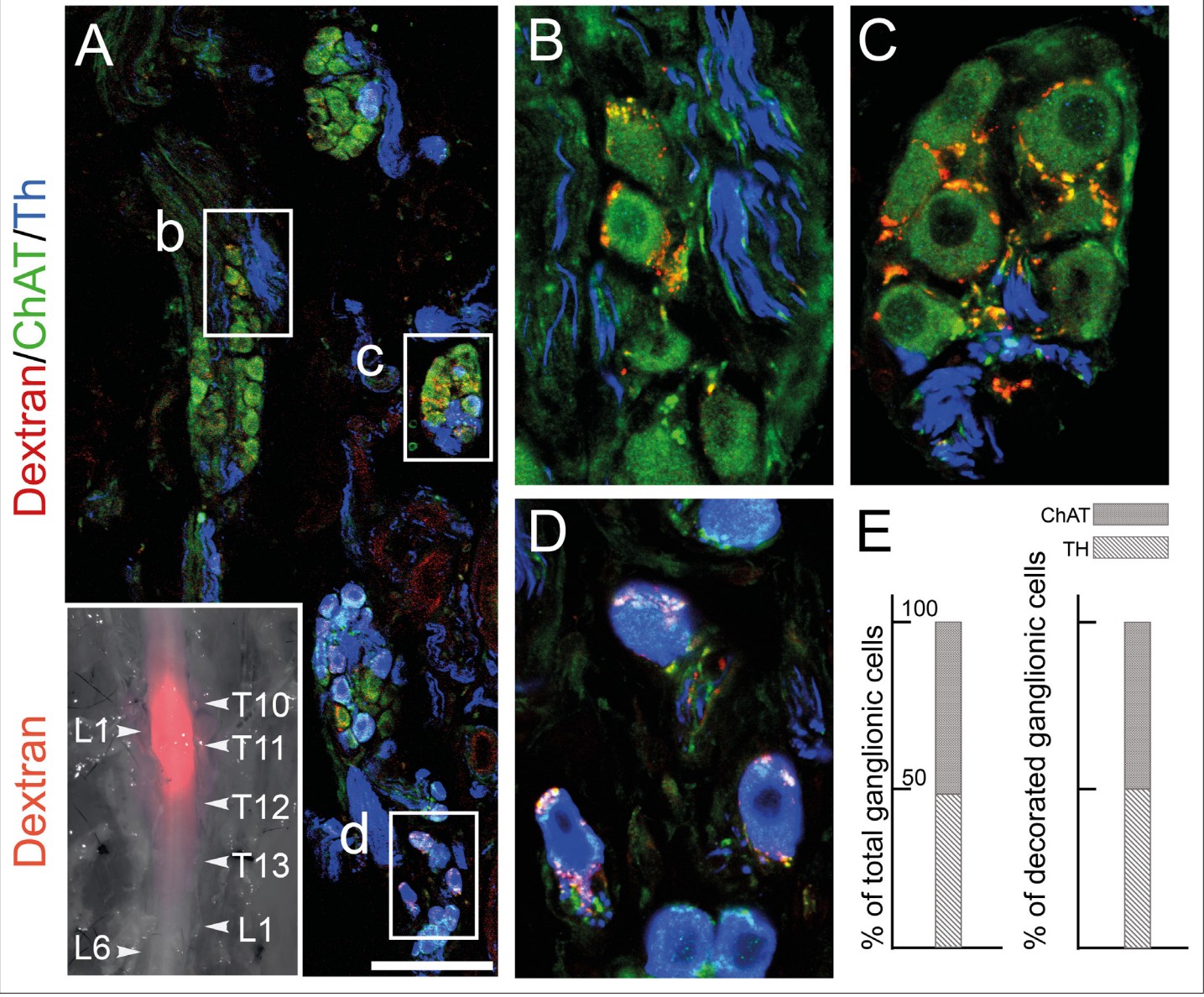

**Figure 3.** The lumbar outflow targets both cholinergic and noradrenergic pelvic ganglionic cells. (**A–E**) Section (A, low magnification, B–D high magnifications of selected regions) through the pelvic ganglion of an adult male mouse stereotactically injected with Dextran at the L1 level of the lumbar spinal cord (inset) and showing dextran filled boutons decorating both choline acetyltransferase (CHAT)+ (**B–C**) and tyrosine hydroxylase (TH)+ cells (**D**). Whether they are filled by Dextran or not, cholinergic boutons (green), presumably from spinal preganglionics (lumbar or sacral), are present on most cells. In the inset, levels of the vertebral column are indicated on the right, levels of the spinal cord on the left. (**E**) Quantification of TH and CHAT cells among total or bouton-decorated ganglionic cells. CHAT+ cells represent 51% of total cells and 50% of decorated cells (for a total of 3186 counted cells, among which 529 decorated cells, on 48 sections in four mice). Scale bar in A: 100μm.

## Discussion

The notion that the pelvic ganglion is a caudal elaboration of the sympathetic ganglionic chains echoes the situation of its preganglionic neurons, in the lumbar and sacral spinal cord. We showed that lumbar and sacral preganglionics are indistinguishable by several criteria, including the expression state of six transcription factors (*Phox2a*, *Phox2b*, *Tbx3*, *Tbx2*, *Tbx20*, *FoxP1*) and their dependency on the bHLH gene *Olig2*, in addition to their well-known location (the intermediate lateral column) and the ventral exit point of their axon (*Espinosa-Medina et al., 2016*). Recent surveys of spinal preganglionic neurons by single-cell transcriptomics (*Blum et al., 2021*; *Alkaslasi et al., 2021*) discovered many subtypes, most of them evenly distributed from thoracic to sacral levels, and a few enriched at,

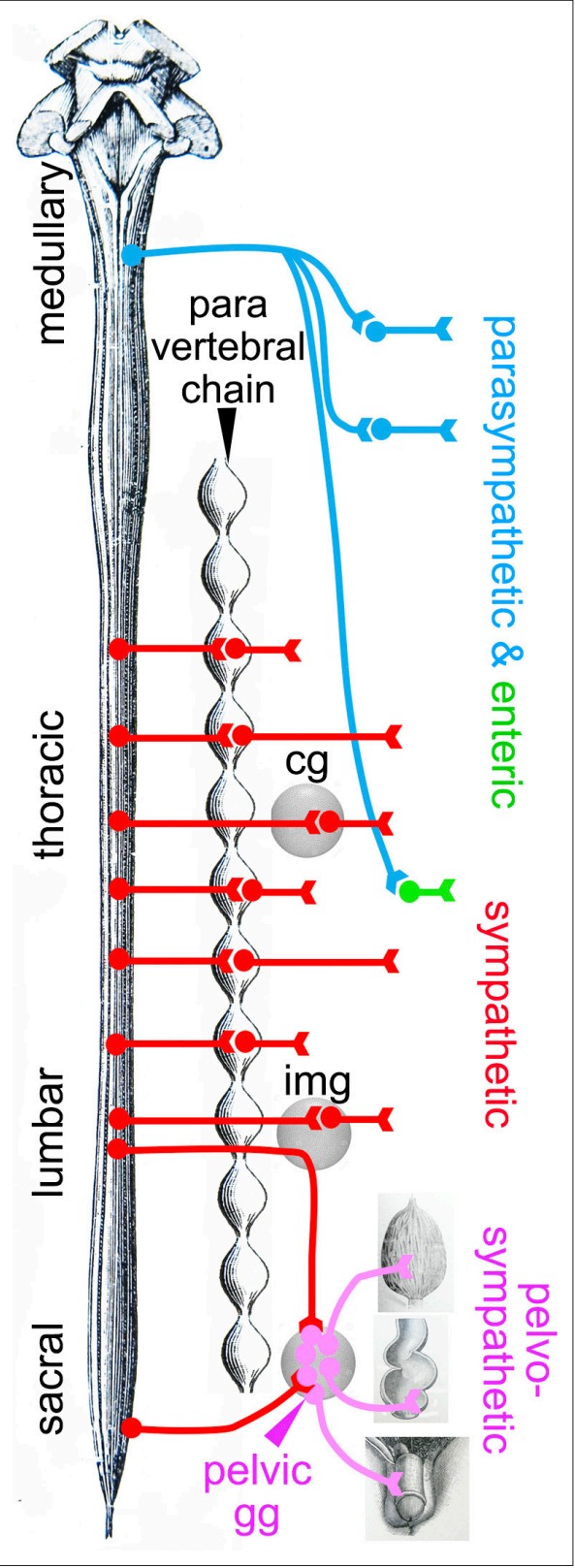

**Figure 4.** Deployment of the divisions of the autonomic nervous system on the rostro-caudal axis. Cg, celiac ganglion; img, inferior mesenteric ganglion; pelvic gg, pelvic ganglion. Only the target organs of the pelvo-sympathetic pathway are represented. The adrenal medulla is omitted. The pelvic ganglion is shown with its lumbar input (through the hypogastric nerve) and sacral input (through the pelvic nerve).

or specific to the sacral level, confirming that the sacral preganglionic neurons are sympathetic, yet represent a caudal modification of the thoraco-lumbar intermediate lateral column.

In conclusion, the genetic signatures of neither pre- nor post-ganglionic neurons of the sacral autonomic outflow support its century-old assignment to the parasympathetic division of the visceral nervous system. Importantly, we have argued (*Espinosa-Medina et al., 2018*) that the supposed parasympathetic identity was unnecessary at best to understand the physiology of the pelvis, at worst incongruent with it, at least concerning sex organs and the bladder. The antagonism between an anti-erectile lumbar and a pro-erectile sacral autonomic pathways on the blood vessels of the external genitals – the key argument that Langley advanced for making the sacral pathway parasympathetic, even before he coined the term (*Langley, 1899*) – was repeatedly challenged by the evidence of a lumbo-sacral pro-erectile synergy (reviewed in *Espinosa-Medina et al., 2018*). And experimental support for a lumbo-sacral antagonism on the bladder is scant, despite common perception from textbooks and reviews (reviewed in *Espinosa-Medina et al., 2018*). It is thus remarkable that the cell type-based anatomy that we uncover in no way contradicts the physiology. There is no need to posit that the sympathetic-like neurons we describe in the pelvic region display a type of physiology common with that of parasympathetic ones in the head or thorax, which would deserve a common name. The classical parasympathetic label is thus best dropped and replaced by the genetic (i.e. embryological and evolutionary) notion of a sympathetic one, modified to suit the unique demands of pelvic physiology. The division of the autonomic nervous system that encompasses the pelvic ganglion and both its lumbar and sacral afferents (which could be termed 'pelvo-sympathetic' and includes whatever antagonistic pathways operate in the region – e.g. on genital blood vessels) is now ripe for finer grain analysis, anatomical and physiological, at the level of neuron types, defined by their gene expression patterns.

# Materials and methods

## Key resources table

| Reagent type (species) or resource | Designation | Source or reference | Identifiers | Additional information |
|---|---|---|---|---|
| Genetic reagent (*Mus musculus*) | Phox2b::Cre mouse line | *D'Autréaux et al., 2011* | | BAC transgenic line expressing Cre under the control of the Phox2b promoter |
| Genetic reagent (*Mus musculus*) | Rosa$^{lox-stop-lox-tdTomato}$ (Rosa$^{tdT}$) mouse line | *Madisen et al., 2010* | | Knock-in line expressing the reporter gene tdTomato from the Rosa locus in a Cre-dependent manner |
| Antibody (primary) | α-Phox2b rabbit polyclonal | *Pattyn et al., 1997* | | IHC and IF (1:500) |
| Antibody (primary) | α-TH rabbit polyclonal | Invitrogen | OPA1-04050 | IF (1:1000) |
| Antibody (primary) | α-Choline acetyltransferase (ChAT) goat polyclonal | Thermo Fisher | PA1-9027 | IF (1:100) |
| Antibodies (secondary) | α-Rabbit PK goat polyclonal | Vector Laboratories | PK-4005 | IHC (1:200) |
| Antibodies (secondary) | Anti-goat 647 donkey polyclonal | Thermo Fisher | A-21447 | IF (1:500) |
| Antibodies (secondary) | Anti-rabbit 488 donkey polyclonal | Thermo Fisher | A-21206 | IF (1:500) |
| Antibodies (secondary) | α-Rabbit Cy3 donkey polyclonal | Jackson | 711-165-152 | IF (1:500) |
| Recombinant DNA reagent | Ebf3 (plasmid) | Gift of S Garel | | |
| Recombinant DNA reagent | Gata3 (plasmid) | Gift of JD Engel | | |
| Recombinant DNA reagent | Hand1 (plasmid) | Gift of P Cserjesi | | |
| Recombinant DNA reagent | Hmx2 (plasmid) | Gift of EE Turner | | |

*Continued on next page*

*Continued*

| Reagent type (species) or resource | Designation | Source or reference | Identifiers | Additional information |
|---|---|---|---|---|
| Recombinant DNA reagent | Hmx3 (plasmid) | Gift of S Mansour | | |
| Recombinant DNA reagent | Islet1 (plasmid) | *Tiveron et al., 1996* (10.1523/JNEUROSCI.16-23-07649.1996) | | |
| Recombinant DNA reagent | Tbx20 (plasmid) | *Dufour et al., 2006* (10.1073/pnas.0600805103) | | |
| Recombinant DNA reagent | Sst (plasmid) | Clone Image ID #4981984 | | |
| Sequence-based reagent | Dlx5_F | This paper | PCR primers | 5' -GACGCAAA CACAGGTGAAAATCTGG-3' |
| Sequence-based reagent | Dlx5_R | This paper | PCR primers | 5'-GGGCGGGGC TCTCTGAAATG-3' |
| Sequence-based reagent | Gata2_F | This paper | PCR primers | 5'-TTGTGTTCTT GGGGTCCTTC-3' |
| Sequence-based reagent | Gata2_R | This paper | PCR primers | 5'-GCTTCTGTGG CAACGTACAA-3' |
| Sequence-based reagent | Hmx1_F | This paper | PCR primers | 5'-CGTTCGCCAC TATCCAAACGGG-3' |
| Sequence-based reagent | Hmx1_R | This paper | PCR primers | 5'-TGTCAGGACT TAGACCACCTCCG-3' |
| Sequence-based reagent | Ntn1_F | This paper | PCR primers | 5'-CTTCCTCACC GACCTCAATAAC-3' |
| Sequence-based reagent | Ntn1_R | This paper | PCR primers | 5'-GCGATTTAG GTGACACTATA GTTGTGCCTACAGTCACACAC C-3' |
| Sequence-based reagent | Syt6_F | This paper | PCR primers | 5'-GTGGTCTTCT TGTCCCGTGT-3' |
| Sequence-based reagent | Syt6_R | This paper | PCR primers | 5'-CATGTGCTTA CAGGGTGTGG-3' |
| Sequence-based reagent | Zbtb16_F | This paper | PCR primers | 5'-ATGAAAACAT ACGGGTGTGAA-3' |
| Sequence-based reagent | Zbtb16_R | This paper | PCR primers | 5'-CCAAGGCCAA GTAACTATCAGG-3' |
| Chemical compound, drug | Tetramethyl-rhodamine-conjugated dextran | Thermo Fisher | D3308 | For tracing experiments |
| Chemical compound, drug | NBT-BCIP solution | Sigma | B1911 | For ISH experiments |
| Chemical compound, drug | 3,3'-Diaminobenzidine (DAB) | Sigma | D12384 | For IHC experiments |
| Software, algorithm | Cell Ranger software | 10x Genomics | | 6.0.1 |

## Mice

Phox2b::Cre (*D'Autréaux et al., 2011*): a BAC transgenic line expressing *Cre* under the control of the *Phox2b* promoter.

Rosa$^{lox-stop-lox-tdTomato}$ (Rosa$^{tdT}$) (*Madisen et al., 2010*): Knock-in line expressing the reporter gene *tdTomato* from the *Rosa* locus in a Cre-dependent manner.

## Obtainment of ganglionic cells

Sphenopalatine, stellate, coeliac, lumbar, and pelvic ganglia were dissected from *Phox2bCre;Rosa$^{tdT}$* P5 pups representing both sexes and placed in artificial cerebrospinal fluid oxygenated with carbogen (4°C). Fat tissue was carefully removed and nerves emanating from the ganglia were cut. Ganglia were transferred into a 1.5 ml Eppendorf tube containing 1 ml PBS-glucose (1 mg/ml glucose), 20 μl papain solution (Worthington LS003126; 25.4 units/mg) and 20 μl DNAse (2 mg/ml in PBS) and incubated

at 37°C for 15 min. The ganglia were collected by centrifugation (300×$g$, 1 min), the supernatant replaced by 1 ml PBS-glucose supplemented with 50 µL Collagenase/Dispase (Worthington CLS-1 345 U/mg; Dispase II Roche 1.2 U/mg; 80 mg Collagenase and 92 mg Dispase II dissolved in 1 ml PBS-glucose) and 20 µl DNAse solution and the ganglia incubated at 37°C for 8 min. After collecting the ganglia by centrifugation for 3 min at 300×$g$ they were dissociated in 1 ml PBS-glucose supplemented with 0.04% bovine serum albumin (BSA) and DNAse (20 µl) by trituration, using a fire-polished, siliconized Pasteur pipette. The cell suspension was then filtered through a 40 µm cell strainer. To eliminate cell debris, the cell suspension was centrifuged through a density step gradient by overlaying the cell suspension onto 1 ml OptiPrep solution (80 µl OptiPrep [Sigma], 900 µl PBS supplemented with 0.04%BSA, 20 µl DNAse) for 15 min, 100×$g$ at 5°C. After removal of the supernatant from the soft cell pellet, the cells were suspended in 100 µl PBS-glucose/0.04% BSA and collected again in a 500 µl Eppendorf tube for 15 min, 100×$g$ at 5°C. The supernatant was carefully removed (under control by fluorescence microscope). After addition of 40 µl PBS-glucose/0.04% BSA, the cell density was adjusted to 1000 cells/µl and transferred to the 10x Genomics platform.

## Library construction and sequencing

Single-cell RNA sequencing was performed in two separate experimental rounds, one for the stellate, sphenopalatine, and pelvic ganglia (pelvic_1) performed at the École normale supérieure GenomiqueENS core facility (Paris, France), and one for the celiac, lumbar and pelvic ganglia (pelvic_2) performed at the ICGex NGS platform of the Institut Curie (Paris, France). Cellular suspensions (10,000 cells for the first round, 5300 cells for the second) were loaded on a 10x Chromium instrument (10x Genomics) to generate single-cell GEMs (5000 for the first round, 3000 for the second). Single-cell RNA sequencing libraries were prepared using Chromium Single Cell 3' Reagent Kit (v2 for the first round, v3 for the second) (10x Genomics) according to the manufacturer's protocol based on the 10x GEMCode proprietary technology. Briefly, the initial step consisted in performing an emulsion where individual cells were isolated into droplets together with gel beads coated with unique primers bearing 10x cell barcodes, UMI (unique molecular identifiers) and poly(dT) sequences. Reverse transcription reactions were applied to generate barcoded full-length cDNA followed by disruption of the emulsions using the recovery agent and the cDNA was cleaned up with DynaBeads MyOne Silane Beads (Thermo Fisher Scientific). Bulk cDNA was amplified using a GeneAmp PCR System 9700 with 96-Well Gold Sample Block Module (Applied Biosystems) (98°C for 3 min; 12 cycles: 98°C for 15 s, 63°C for 20 s, and 72°C for 1 min; 72°C for 1 min; held at 4°C). The amplified cDNA product was cleaned up with the SPRIselect Reagent Kit (Beckman Coulter). Indexed sequencing libraries were constructed using the reagents from the Chromium Single Cell 3' Reagent Kit v3, in several steps: (1) fragmentation, end repair, and A-tailing; (2) size selection with SPRI select; (3) adaptor ligation; (4) post-ligation cleanup with SPRI select; (5) sample index PCR and cleanup with SPRI select beads (with 12–14 PCR cycles depending on the samples). Individual library quantification and quality assessment were performed using the Qubit fluorometric assay (Invitrogen) with the dsDNA HS (High Sensitivity) Assay Kit and a Bioanalyzer Agilent 2100 using a High Sensitivity DNA chip (Agilent Genomics). Indexed libraries were then equimolarly pooled and quantified by qPCR using the KAPA library quantification kit (Roche). Sequencing was performed on a NextSeq 500 device (Illumina) for the first round and a NovaSeq 6000 (Illumina) for the second, targeting around 400M clusters per sample and using paired-end (26/57 bp for the first round, 28×91 bp for the second).

## Bioinformatic analysis

For each of the six samples (pelvic_1, stellate, sphenopalatine, pelvic_2, coeliac, and lumbar) we performed demultiplexing, barcode processing, and gene counting using the Cell Ranger software (v. 6.0.1). The 'filtered_feature_bc_matrix' files (uploaded at https://www.ncbi.nlm.nih.gov/geo/query/acc.cgi?acc=GSE232789) were used as our starting points for defining cells. For each dataset, only droplets that expressed more than 1500 genes, less than 11,000 genes, and a percentage of mitochondrial genes below 15% were retained. This resulted in the selection of 4404 pelvic_1 cells, 7225 stellate cells, 4630 sphenopalatine cells, 1643 pelvic_2 cells, 1428 coeliac, and 2120 lumbar cells.

We used Seurat version 3 (*Stuart et al., 2019*) to read, manipulate, assemble, and normalize (*Hafemeister and Satija, 2019*) the datasets. Specifically, we concatenated cells from all six datasets and normalized the data using the sctransform (SCT) method for which we fitted a Gamma-Poisson

generalized linear model ('glmGamPoi' option). Using the Seurat framework, we then performed a PCA on the normalized dataset.

Next, we integrated all six datasets using Batchelor (*Haghverdi et al., 2018*) (a strategy for batch correction based on the detection of mutual nearest neighbors), and we visualized all cells, including neurons, in two dimensions using the UMAP method (*Becht et al., 2018*) on the first 50 components from the PCA.

We selected neurons based on their mean expression of a set of neuronal marker genes (*Stmn2*, *Stmn3*, *Gap43*, and *Tubb3*). Specifically, cells were classified as neurons if their mean marker SCT-normalized gene expression exceeded a threshold of 3, that best separated the bimodal distribution of the higher and lower values of the mean expression of neuronal markers. Likewise, we excluded glial-containing doublets using the following markers: *Plp1*, *Ttyh1*, *Fabp7*, *Cryab*, and *Mal*. The number of neurons thus selected was 862 for Pelvic_1, 2689 for the stellate, 1857 for the sphenopalatine, 361 for Pelvic_2, 236 for celiac, and 925 for the lumbar chain.

We then clustered pelvic neurons based on the first two components from the UMAP generated from the 50 batch-corrected components, using the graph-based clustering method Louvain (*Blondel et al., 2008*) implemented in the Seurat framework, with a resolution of 0.3. This procedure defined 24 clusters and split the pelvic ganglion into four clusters 1, 7, 15, and 19 (*Figure 1—figure supplement 1B*), that we renamed P1, P2, P3, P4 (*Figure 1A*). The efficiency of batch correction is attested by the equal contribution from both batches of pelvic ganglia to all four defined pelvic clusters (P1–4) (*Figure 1—figure supplement 1A*), and the comparable expression level of the top 100 genes across both pelvic batches in the violin plots (*Supplementary file 2*).

To search in an unbiased manner for gene expression similarities between pelvic neurons and either sympathetic or parasympathetic ones, we systematically compared the expression of every gene of the dataset across all possible splits among ganglia, i.e., between every subset of ganglia ('subset_1'), and its complementary subset ('subset_2'). Because we treated each of the 4 pelvic clusters as a ganglion, there were $2^8=256$ such splits (2 of them, defined by 'all_8_groups vs none' and 'none vs all_8_groups', being meaningless). For every split and every gene, we devised a score that would best reflect how higher the proportion is of cells expressing the gene (i.e. one read of the gene or more) in subset_1 than in subset_2. We used a metric based on the product of the proportions of cells expressing the gene in each ganglion of subset_1 or subset_2 (rather than throughout subset_1 or subset_2), to give an equal weight to each ganglion (irrespective of its size), and to penalize splits that contain outliers (i.e. ganglia that contain much fewer cells positive for that gene than other ganglia in subset_1, or ganglia that contain much more cells positive for that gene than other ganglia in subset_2).

Specifically, for a given split and a given gene, the score (here defined by the variable score, in the R programming language) is defined as:

$$score = \sum_{i \in S_1} \frac{1}{n_{S_1}} \cdot log_2 \left( -1 + eps_{pos} + \frac{numCellsExpr\,(i)}{(n_{cells}\,(i))} \right) - \sum_{j \in S_2} \frac{1}{n_{S_2}} \cdot log_2 \left( eps_{nega} + \frac{numCellsExpr\,(j)}{(n_{cells}\,(j))} \right)$$

where:

- $S_1$ and $S_2$ are the ganglion subsets subset_1 and subset_2 as described above.
- $n_{S_1}$ and $n_{S_2}$ are the number of ganglion clusters that belong to subset_1 and subset_2. $\frac{numCellsExpr(i)}{(n_{cells}(i))}$ and $\frac{numCellsExpr(j)}{(n_{cells}(j))}$ are the proportion of cells from ganglion or pelvic cluster *i* and *j* (*i* belonging to subset_1 and *j* to subset_2) which express the given gene.
- $eps_{pos}=0.9$ and $eps_{nega}=0.02$ are parameters controlling a tradeoff between favoring genes expressed in subset_1 and penalizing genes expressed in subset_2.

  Note that when $eps_{pos}=0.9$ (the parameter value used here) the quantity $\left( -1 + eps_{pos} + \frac{numCellsExpr(i)}{(n_{cells}(i))} \right)$ can be negative, in which case the expression becomes equal to -Inf and the gene gets the lowest possible score for that given split. This means that with these settings all dichotomies involving a ganglion or pelvic cluster in which less than 10% of cells express the gene (p=0.1) will get the minimal possible score.

## In vivo gene expression

### Tissue preparation
Embryo sections: E16.5 WT embryos were freshly dissected, washed with 1× PBS, and fixed overnight in 4% PFA at 4°C. Tissues were washed a few times in 1× PBS and treated in 15% sucrose to

cryopreserve the tissues for frozen tissue sections. Tissues were then embedded in the optimal cutting temperature compound and snap-frozen. Tissues were cut at 14 µm on a cryostat. Tissue sections were stored at –80°C until use for staining.

## In situ hybridization

Frozen tissue sections were washed in 5× SSC buffer 15 min at room temperature (RT) and treated in the pre-hybridization solution (50% formamide, 5× SSC buffer, and 40 µg/ml Herring sperm DNA in $H_2O$) for 1 hr at 60°C. Then, slides were put in the hybridization solution (50% formamide, 5× SSC buffer, 5× Denhardt's, 500 µg/ml Herring sperm DNA, 250 µg/ml yeast RNA, and 1 mM DTT in $H_2O$) containing the probe (100 ng/ml), at least two overnights at 60°C. Slides were washed two times in 5× SSC buffer (5 min) and two times in 0.2× SSC buffer (30 min) at 70°C, and then, three times in TBS at RT (10 min). Then, tissues were put in the blocking solution (TBS+10% FCS) for 1 hr in the dark, at RT and in humid atmosphere (250 µl/slide) and incubated 1 hr with the primary antibody (anti-DIG) diluted 1/200 in blocking solution (250 µl/slide). Then, slides were washed again three times in TBS (10 min) and treated 5 min in the AP buffer solution (100 nM Tris pH 9.5, 50 nM $MgCl_2$, and 100 nM NaCl in $H_2O$). The revelation was made by the NBT-BCIP solution (Sigma) in the dark (250 µl/slide). The reaction was stopped in PBS-Tween (PBST).

## Immunohistochemistry

After in situ hybridization, frozen tissues were fixed 15 min in 4% PFA at RT and washed in PBST. Then, tissues were incubated in PBST+10% FCS in the dark for 1 hr (500 µl/slide, without coverslip) and incubated overnight with the primary antibody in the same solution at 4°C (250 µl/slide). Slides were washed in PBST three times (10 min) and incubated for 2 hr at RT with the secondary antibody in the same solution again. Tissues were washed three times in PBST and then incubated at RT with avidin/biotin solution diluted 1/100 in 1× PBS. Then, tissues were washed three times and the revelation was made in 3,3'-diaminobenzidine (DAB) solution containing DAB and urea in $H_2O$ (Sigma). The reaction was stopped in PBST, and slides were washed in PBST and in ultra-pure $H_2O$. Slides were mounted in Aquatex mounting medium (Merck).

## Imaging

Tissues processed by an in situ hybridization and immunohistochemistry were photographed on a Leica bright-field microscope with a 40× oil immersion objective. Images were then treated by Photoshop v. 24.1.0.

## Anterograde tracing

### Tracer injections

Surgeries were conducted under aseptic conditions using a small animal digital stereotaxic instrument (David Kopf Instruments). Male, C57bl mice (2–4 months of age) were anesthetized with isoflurane (3.5% at 1 l/min for induction and 2–3% at 0.3 l/min for maintenance). Carprofen (0.5 mg/kg) was administered subcutaneously for analgesia before surgery. A feedback-controlled heating pad was used to maintain the animal temperature at 36°C. Anesthetized animals were shaved along the spine and placed in a stereotaxic frame. The skin overlying the spine was sterilized with alternating scrubs of Vetadine and 80% (wt/vol) ethanol and a 100 µl injection of lidocaine (2%) was made subcutaneously along the spine before an incision was made from the iliac crest to the T9 vertebral spinous process. The T10 and T11 spinous processes were identified by counting rostrally from the L6 spinous process (identified relative to the iliac crest). Two small rostro-caudal incisions were made on each side of the vertebral column through the superficial-most layer of muscle. Parallel clamps were then placed within these incisions until firm contact was made with the transverse process of the T11 vertebra. The T11 vertebra was then raised upward via the clamps until no respiratory movement was observed. The muscles connecting the T10 spinous process to transverse processes of caudal vertebra were dissected to expose the T10/T11 intervertebral space. A lateral incision was made across the dura to expose the underlying L1 spinal segment and 150 nl injections of 4% lysine fixable, tetramethyl-rhodamine or Alexa-488-conjugated dextran, 3000 MW (Thermo Fisher) were made bilaterally, into the intermediolateral nucleus and intermedio-medial nucleus (0.600 mm lateral, 0.600 mm deep and

0.150 mm lateral and 0.700 mm deep) using a narrow-tapered glass pipette and a Nanoject III injector (Drummond Scientific). Injections were made at 5 nl per second and left in place for 5 min following injection to prevent dextran leakage from the injection site. The pipette was then retracted, and the intervertebral space bathed with sterile physiological saline. A small piece of sterile gel foam hemostat (Pfizer) was then placed within the intervertebral space and the incision overlying the spine was sutured closed.

## Histology

7–14 days after injection the mice were intracardially perfused with cold PBS until exsanguinated and subsequently fixed by perfusion with cold 4% PFA until the carcass became stiff. The bladder, prostate, and pelvic ganglia were immediately dissected as an intact bloc and post-fixed in 4% PFA overnight at 4 °C. The intact spinal column was also dissected out and the dorsal surface of the spinal cord exposed before post-fixation overnight in 4% PFA at 4 °C. The fixed tissues were then rinsed in PBS 3×30 min the following day. The 'bladder block' was then cryopreserved in 15% sucrose (wt/vol) in PBS until non-buoyant. The dorsal aspect of the injection site was imaged in situ using fluorescence stereoscope. The spinal cord was then dissected out and cryopreserved as described above.

The bladder block was sectioned on a cryostat in a sagittal orientation to capture serial sections of the entire pelvic ganglion. Sections were cut at a thickness of 30 μm and collected as a 1 in 4 series on Superfrost Plus glass slides. The spinal cord was sectioned coronally at a thickness of 60 μm as a 1 in 2 series. On slide immunohistochemistry was performed against tyrosine hydroxylase (TH) and choline acetyltransferase (CHAT) for the bladder block sections, primary incubation: 4–6 hr at RT, secondary incubation: 2 hr at RT, with 3×5 min washes in PBS after each incubation. Sections were then mounted with dako fluorescence mounting medium and coverslipped.

## Imaging

Sections were imaged on a Leica Stellaris 5 confocal microscope (Leica microsystems). Images of the pelvic ganglia were captured as Z-stacks with 1 μm interslice distances, with a 20× objective at 2000 mp resolution.

## Counting

All ganglionic cells and cells surrounded by dextran labeled varicosities were identified as either CHAT or TH positive and counted, on a total of 4 animals, 48 sections, and 3186 cells.

## Probes

Primers were designed to amplify by PCR probe templates for the following genes:

| Probe | Forward primer | Reverse primer |
| --- | --- | --- |
| Dlx5 | 5'-GACGCAAACACAGGTGAAAATCTGG-3' | 5'-GGGCGGGGCTCTCTGAAATG-3' |
| Gata2 | 5'-TTGTGTTCTTGGGGTCCTTC-3' | 5'-GCTTCTGTGGCAACGTACAA-3' |
| Hmx1 | 5'-CGTTCGCCACTATCCAAACGGG-3' | 5'-TGTCAGGACTTAGACCACCTCCG-3' |
| Ntn1 | 5'-CTTCCTCACCGACCTCAATAAC-3' | 5'-GCGATTTAGGTGACACTATAGTTGT GCCTACAGTCACACACC-3' |
| Syt6 | 5'-GTGGTCTTCTTGTCCCGTGT-3' | 5'-CATGTGCTTACAGGGTGTGG-3' |
| Zbtb16 | 5'-ATGAAAACATACGGGTGTGAA-3' | 5'-CCAAGGCCAAGTAACTATCAGG-3' |

The PCR fragment were ligated into a pGEM-T Easy Vector System (Promega), transformed into chemically competent cells and sequenced. The other plasmid templates were *Ebf3* (gift of S Garel), *Gata3* (gift of JD Engel), *Hand1* (gift of P Cserjesi), *Hmx2* (gift of EE Turner), *Hmx3* (gift of S Mansour), *Islet1* (*Tiveron et al., 1996*), *Tbx20* (*Dufour et al., 2006*) and *Sst* (Clone Image ID #4981984).

Plasmids were digested by restriction enzymes and purified using a DNA Clean & Concentrator kit (Zymo Research). Antisense probes were synthesized with RNA polymerases and a DIG RNA labeling mix, and purified by the ProbeQuant G-50 micro columns kit (GE Healthcare). Probes were stored at –20°C.

## Antibodies

Primary antibodies were α-PHOX2B rabbit (1:500 or 1:1000, *Pattyn et al., 1997*). α-TH (Invitrogen: OPA1-04050, 1:1000) and α-CHAT (Thermo Fisher: PA1-9027, 1:100).

Secondary antibodies were goat α-rabbit (PK-4005, Vector Laboratories), donkey anti-goat 647 (A-21447, Thermo Fisher), donkey anti-rabbit 488 (A-21206, Thermo Fisher), and donkey α-rabbit Cy3 (711-165-152, Jackson).

## Acknowledgements

We thank Nicolas Narboux-Nême for help with the analysis of *Dlx5* expression, Sonia Garel for the gift of the *Ebf3* probe, Amandine Delecourt and Gwendoline Firmin for help with the mouse colonies. The Brunet laboratory is supported by INSERM, CNRS, ANR-19-CE16-0029-01, ANR-17-CE16-0006-01, FRM EQU202003010297. MA is a recipient of a doctoral fellowship from Labex MemoLife. BD was funded by grant APP2001128 from the National Health & Medical Research Council, Australia. HR has been supported by a grant from the Wilhelm Sander Foundation and from Labex MemoLife. High-throughput sequencing was performed by (1) the GenomiqueENS core facility supported by the France Génomique national infrastructure, funded as part of the 'Investissements d'Avenir' program managed by the Agence Nationale de la Recherche (contract ANR-10-INBS-0009); (2) the ICGex NGS platform of the Institut Curie supported by the grants ANR-10-EQPX-03 (Equipex) and ANR-10-INBS-09–08 (France Génomique Consortium) from the Agence Nationale de la Recherche ('Investissements d'Avenir' program), by the ITMO-Cancer Aviesan (Plan Cancer III) and by the SiRIC-Curie program (SiRIC Grant INCa-DGOS-465 and INCa-DGOS- Inserm_12554). Data management, quality control, and primary analysis were performed by the Bioinformatics platform of the Institut Curie.

## Additional information

### Funding

| Funder | Grant reference number | Author |
|---|---|---|
| Agence Nationale de la Recherche | ANR-19-CE16- 0029-01 | Jean-François Brunet |
| Agence Nationale de la Recherche | ANR-17-CE16-0006-01 | Jean-François Brunet |
| Fondation pour la Recherche Médicale | FRM EQU202003010297 | Jean-François Brunet |
| National Health and Medical Research Council | APP2001128 | Bowen Dempsey |

The funders had no role in study design, data collection and interpretation, or the decision to submit the work for publication.

### Author contributions

Margaux Sivori, Validation, Investigation, Visualization, Writing – review and editing; Bowen Dempsey, Validation, Investigation, Methodology; Zoubida Chettouh, Franck Boismoreau, Maïlys Ayerdi, Annaliese Eymael, Fanny Coulpier, Investigation; Sylvain Baulande, Sonia Lameiras, Methodology; Olivier Delattre, Resources; Hermann Rohrer, Conceptualization, Investigation, Writing – review and editing; Olivier Mirabeau, Data curation, Formal analysis, Methodology, Writing – review and editing; Jean-François Brunet, Conceptualization, Funding acquisition, Writing – original draft, Project administration, Writing – review and editing

### Author ORCIDs

Hermann Rohrer (ID) https://orcid.org/0000-0001-7023-1355
Olivier Mirabeau (ID) http://orcid.org/0000-0002-4048-1385
Jean-François Brunet (ID) https://orcid.org/0000-0002-1985-6103

## Ethics

Breeding of mice, obtainment of mouse embryos and of cells from mouse pups were performed in strict accordance with the recommendations of the French Ministry of Agriculture and have been approved by the Direction Départementale des Services Vétérinaires de Paris. Tracer injection experiments were approved by the Macquarie University Animal Ethics Committee (animal research authority# 2018-024) and conformed to the Australian Code of Practice for the Care and Use of Animals for Scientific Purposes 2013.

Reviewer #1 (Public Review): https://doi.org/10.7554/eLife.91576.3.sa1
Reviewer #2 (Public Review): https://doi.org/10.7554/eLife.91576.3.sa2
Author Response https://doi.org/10.7554/eLife.91576.3.sa3

# Additional files

## Supplementary files

• Supplementary file 1. Table of the 500 highest-scoring genes for each of the 254 possible dichotomizations of ganglia and pelvic clusters (i.e. partitioning the ganglia and pelvic clusters into two mutually exclusive sets, subset_1 and subset_2), ordered by score, and after removal, for each gene, of all dichotomizations below the highest scoring one (resulting in the ranking of 7593 genes). Each row in the table indicates the gene name, the score, a description of the dichotomization (under the field 'group.compar.string', in the format {subset_1}vs{subset_2}), and each ganglion or pelvic cluster, marked with 1 or 0 to indicate that the cluster belongs, respectively, to subset_1 or subset_2.

• Supplementary file 2. Violin plots of the top 100 highest-scoring genes, displaying the SCT-normalized values for each of the eight ganglion or pelvic cluster: stellate, sphenopalatine, pelvic_1, coeliac, lumbar, and pelvic_2 color-coded as indicated.

• Supplementary file 3. Bar plots of the top 100 highest-scoring genes, displaying the proportion of cells (ranging between 0 and 1) in each ganglion or pelvic cluster that express at least one read of a given gene.

• MDAR checklist

## Data availability

The data can be accessed at: https://www.ncbi.nlm.nih.gov/geo/query/acc.cgi?acc=GSE232789.

The following dataset was generated:

| Author(s) | Year | Dataset title | Dataset URL | Database and Identifier |
|---|---|---|---|---|
| Brunet J, Mirabeau O, Rohrer H, Sivori M, Lameiras S, Coulpier F, Baulande S | 2023 | The pelvic ganglion is a divergent outpost of the sympathetic chains | https://www.ncbi.nlm.nih.gov/geo/query/acc.cgi?acc=GSE232789 | NCBI Gene Expression Omnibus, GSE232789 |

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
