## [Editor Report · eLife assessment]

This **useful** study compares gene expression patterns among different autonomic ganglia and will be of interest to developmental neuroscientists and neurophysiologists. The study expands the database of genes expressed by subpopulations of autonomic neurons in ganglia, a key step in decoding their developmental origins and physiological functions. The evidence supporting the alternative view that the pelvic ganglionic neurons are actually modified sympathetic neurons is **incomplete** and may cause confusion, given the enrichment of cholinergic neurons, as well as the large number of molecular and functional differences known to be present between cranial and sacral neurons.

---

## [Referee Report · Reviewer #1 (Public Review)]

In recent years, these investigators have been engaged in a debate regarding the classification of the sacral parasympathetic system as "sympathetic" rather than "parasympathetic," based on shared developmental ontogeny of spinal preganglionic neurons. In this current study, these investigators conducted single-cell RNAseq analyses of four groups of autonomic neurons: paravertebral sympathetic neurons (stellate and lumbar train ganglia), prevertebral sympathetic neurons (coeliac-mesenteric ganglia), rostral parasympathetic ganglia (sphenopalatine ganglia), and the caudal pelvic ganglia (containing traditionally recognized sacral "parasympathetic cholinergic neurons," which the investigators sought to challenge in terms of nomenclature). The authors argued that the pelvic ganglionic neurons shared the expression of more genes with sympathetic ganglia, as opposed to parasympathetic ganglia. Additionally, the pelvic neurons did not express a set of genes observed in the rostral parasympathetic sphenopalatine ganglia. Based on these findings, they claimed that the sacral autonomic system should be considered sympathetic rather than parasympathetic.

However, noradrenergic sympathetic neurons and cholinergic neurons, by the virtue of expressing different neurotransmitters, could have distinct roles. It is true that some cholinergic neurons reside in the sympathetic train ganglia as well, such as those innervating the sweat gland and some vascular systems; in this sense, the pelvic ganglia share some features with sympathetic ganglia, except that the pelvic ganglia contain a much higher percentage of cholinergic neurons compared with sympathetic ganglia. It is much simpler and easier to divide the autonomic nervous system into sympathetic neurons that relieve noradrenaline versus parasympathetic neurons that relieve acetylcholine, and these two systems often act in antagonistic manners, even though in some cases, these two systems can work synergistically. As such, it is not justified to claim that "pelvic organs receive no parasympathetic innervation".

---

## [Referee Report · Reviewer #2 (Public Review)]

Summary:

Recent advances in single cell profiling of gene expression (RNA) permit the analysis of specialized cell types, an approach that has great value in the nervous system which is characterized by prodigious neuronal diversity. The novel data in this study focus primarily on genetic profiling to compare autonomic neurons from ganglia associated with the cranial parasympathetic outflow (sphenopalatine also known as pteropalatine), the thoraco-lumbar sympathetic outflow (stellate, coeliac) and the sacral parasympathetic outflow (pelvic). Using statistical methods to reduce the dimensionality of the data and map gene expression, the authors provide interesting evidence that cranial parasympathetic and sacral sympathetic ganglia differ from each other and from sympathetic ganglia (Figures 1, S1 - S4). The authors interpret the mapping analysis as evidence that the cranial and sacral outflows differ so that calling them both parasympathetic is unjustified. Based on anatomical localization of markers (Figure 2 ) (mainly transcription factors) the authors show a similarity between the sympathetic and pelvic ganglion. In Figure 3 they present evidence that some pelvic ganglionic neurons are dually innervated by sympathetic preganglionic neurons and sacral preganglionic neurons. These observations are interpreted to mean that the pelvic ganglion is not parasympathetic, but rather a modified sympathetic ganglion - hence the title of the manuscript.

Strengths:

The extensive use of single cell profiling in this work is both interesting and exciting. Although still in its early stages, it holds promise for a deepened understanding of autonomic development and function. As noted in the introduction, this study extends previous work by Professor Brunet and his associates.

Weaknesses:

This work further documents differences between the cranial and sacral parasympathetic outflows that have been known since the time of Langley - 100 years ago. The approach taken by Brunet et al. has focused on late neonatal and early postnatal development, a time when autonomic function is still maturing. In addition, the sphenopalatine and other cranial ganglia develop from placodes and the neural crest, while sympathetic and sacral ganglia develop from the neural crest alone. How then do genetic programs specifying brainstem and spinal development differ and how can this account for kinship that Brunet documents between spinal and sacral ganglia? One feature that seems to set the pelvic ganglion apart is the mixture of 'sympathetic' and 'parasympthetic' ganglion cells and the convergence of preganglionic sympathetic and parasympathetic synapses on individual ganglion cells (Figure 3). This unusual organization has been reported before using microelectrode recordings (see Crowcroft and Szurszewski, J Physiol (1971) and Janig and McLachlan, Physiol Rev (1987)). Anatomical evidence of convergence in the pelvic ganglion has been reported by Keast, Neuroscience (1995). It should also be noted that the anatomy of the pelvic ganglion in male rodents is unique. Unlike other species where the ganglion forms a distributed plexus of mini-ganglia, in male rodents the ganglion coalesces into one structure that is easier to find and study. Interestingly the image in Figure 3A appears to show a clustering of Chat-positive and Th-positive neurons. Does this result from the developmental fusion of mini ganglia having distinct sympathetic and parasympathetic origins. In addition, Brunet et al dismiss the cholinergic and noradrenergic phenotypes as a basis for defining parasympathetic and parasympathetic neurons. However, see the bottom of Figure S4 and further counterarguments in Horn (Clin Auton Res (2018)). What then about neuropeptides, whose expression pattern is incompatible with the revised nomenclature proposed by Brunet et al.? Figure 1B indicates that VIP is expressed by sacral and cranial ganglion cells, but not thoracolumbar ganglion cells. The authors do not mention neuropeptide Y (NPY). The immunocytochemistry literature indicates that NPY is expressed by a large subpopulation of sympathetic neurons but never by sacral or cranial parasympathetic neurons.

The title of this paper is misleading because it implies a conclusion that is not adequately supported by the data and that is difficult for a general reader to parse. Independent assessments by two referees both agreed on title's problematic message. If one can get beyond the title, then the paper does contain data that is of interest. The authors compared single cell gene expression in neurons from the cranial sphenopalatine ganglion, paravertebral chain ganglia (stellate and lumbar), the prevertebral coeliac ganglion and the bladder ganglion. The cranial and pelvic ganglia are parasympathetic, while the paravertebral and prevertebral ganglia are sympathetic. The gene expression data identified differences between the cranial, sympathetic, and pelvic ganglia. Based primarily on this finding the authors concluded that the sacral bladder ganglion is not parasympathetic. Since some genes suggest a kinship between the pelvic and sympathetic neurons, the authors conclude that the pelvic neurons are pelvo-sympathetic - hence the title. This nomenclature does little to improve understanding of the autonomic motor system and it ignores important anatomical and functional properties that underlie existing definitions of the sympathetic and parasympathetic systems. The idea that the cranial and sacral autonomic outflows have some differences is not new (see for example Nilsson, 1983 and Janig, 2022). Since many of the genes identified in the present study are HOX genes and other transcription factors that specify the rostro-caudal axis during development, it is also not surprising that these genes suggest a kinship between sacral parasympathetic neurons and sympathetic neurons, all of which derive from the neural crest and are supplied by the spinal cord. The different profile of cranial parasympathetic neurons is also not surprising given that they derive from a mixture of placodal and neural crest progenitors and are supplied by the brainstem. see my previous comments for anatomical and functional criteria that further support the existing nomenclature for the sympathetic and parasympathetic motor systems.

---

## [Author Response]

**Author responses to the original review:**

The data we produce are not criticized as such and thus, do not require revision; the criticisms concern our interpretation of them. General themes of the reviews are that (i) genetic signatures do not matter for defining neuronal types (here sympathetic versus parasympathetic); (ii) that a cholinergic postganglionic autonomic neuron must be parasympathetic; and (iii) that some physiology of the pelvic region would deserve the label “parasympathetic”. We answered the latter argument in (Espinosa-Medina et al., 2018) to which we refer the interested reader; and we fully disagree with the first two. Of note, part of the last sentence of the eLife assessment is misleading and does not reflect the referees’ comments. Our paper analyses genetic differences between the cranial and sacral outflow and uses them to argue that they cannot be both parasympathetic. The eLife assessment acknowledges the “genetic differences” but concludes that, somehow, they don’t detract from a common parasympathetic identity. We take issue with this paradox, of course, but it is coherent with the referee’s comments. On the other hand, the eLife assessment alone pushes the paradox one step further by stating that “functional differences” between the cranial and sacral outflows can’t either prevent them from being both parasympathetic. We would also object to this, but the only “functional differences” used by the referees to dismiss our diagnostic of a sympathetic-like character (rather than parasympathetic) for the sacral outflow are between noradrenergic and cholinergic, and between sympathetic and parasympathetic (and we also disagree with those, see above, and below) —not between cranial and sacral.

We will thus use the opportunity offered by eLife to keep the paper as it is (with a few minor stylistic changes). We respond below to the referees’ detailed remarks and hope that the publication, as per eLife new model, of the paper, the referees’ comments and our response will help move the field forward.

**Public review by Referee #1**

“Consistently, the P3 cluster of neurons is located close to sympathetic neuron clusters on the map, echoing the conventional understanding that the pelvic ganglia are mixed, containing both sympathetic and parasympathetic neurons”.

The greater closeness of P3 than of P1/2/4 to the sympathetic cluster can be used to judge P1/2/4 less sympathetic than P3 (and more… something else), but not more parasympathetic. There is no echo of the “conventional understanding” here.

“A closer look at the expression showed that some genes are expressed at higher levels in sympathetic neurons and in P2 cluster neurons ” [We assume that the referee means “in sympathetic neurons and in P3 cluster neurons”] but much weaker in P1, P2, and P4 neurons such as Islet1 and GATA2, and the opposite is true for SST. Another set of genes is expressed weakly across clusters, like HoxC6, HoxD4, GM30648, SHISA9, and TBX20.

These statements are inaccurate; On the one hand, the classification is not based on impression by visual inspection of the heatmap, but by calculations, using thresholds. Admittedly, the thresholds have an arbitrary aspect, but the referee can verify (by eye inspection of heatmap) that genes which we calculate as being at “higher levels in sympathetic neurons and in P3 cluster neurons, but much weaker in P1, P2, and P4 neurons” or vice versa, i.e. noradrenergic or cholinergic neurons (genes from groups V and VI, respectively), have a much bigger difference than those cited by the referee, indeed are quasi-absent from the weaker clusters or ganglia. In addition, even by subjective eye inspection:

Islet is equally expressed in P4 and sympathetics.

SST is equally expressed in P1 and sympathetics.

Tbx20 is equally expressed in P2 and sympathetics.

HoxC6, HoxD4, GM30648, SHISA9 are equally expressed in all clusters and all sympathetic ganglia.

“Since the pelvic ganglia are in a caudal body part, it is not surprising to have genes expressed in pelvic ganglia, but not in rostral sphenopalatine ganglia, and vice versa (to have genes expressed in sphenopalatine ganglia, but not in pelvic ganglia), according to well recognized rostro-caudal body patterning, such as nested expression of hox genes.”

We do not simply show “genes expressed in pelvic ganglia, but not in rostral sphenopalatine ganglia, and vice versa”, i.e. a genetic distance between pelvic and sphenopalatine, but many genes expressed in all pelvic cells and sympathetic ones, i.e. a genetic proximity between pelvic and sympathetic. This situation can be deemed “unsurprising”, but it can only be used to question the parasympathetic nature of pelvic cells (as we do), or considered irrelevant (as the referee does, because genes would not define cell types, see our response to an equivalent stance by Referee#2). Concerning Hox genes, we do take them into account, and speculate in the discussion that their nested expression is key to the structure of the autonomic nervous system, including its division into sympathetic and parasympathetic outflows.

It is much simpler and easier to divide the autonomic nervous system into sympathetic neurons that release noradrenaline versus parasympathetic neurons that release acetylcholine, and these two systems often act in antagonistic manners, though in some cases, these two systems can work synergistically. It also does not matter whether or not pelvic cholinergic neurons could receive inputs from thoracic-lumbar preganglionic neurons (PGNs), not just sacral PGNs; such occurrence only represents a minor revision of the anatomy. In fact, it makes much more sense to call those cholinergic neurons located in the sympathetic chain ganglia parasympathetic.

This “minor revision of the anatomy” would make spinal preganglionic neurons which are universally considered sympathetic (in the thoraco-lumbar chord), synapse onto large numbers of parasympathetic neurons (in the paravertebral chains for sweat glands and periosteum, and in the pelvic ganglion), robbing these terms of any meaning.

Thus, from the functionality point of view, it is not justified to claim that "pelvic organs receive no parasympathetic innervation".

There never was any general or rigorous functional definition of the sympathetic and parasympathetic nervous systems — it is striking, almost ironic, that Langley, creator of the term parasympathetic and the ultimate physiologist, provides an exclusively anatomic definition in his Autonomic Nervous System, Part I. Hence, our definition cannot clash with any “functionality point of view”. In fact, as we briefly say in the discussion and explore in (Espinosa-Medina et al., 2018), it is the “sacral parasympathetic” paradigm which is unjustified from a functionality point of view, for implying a functional antagonism across the lumbo-sacral gap, which has been disproven repeatedly. It remains to be determined which neurons are antagonistic to which on the blood vessels of the external genitals; antagonism within one division of the autonomic nervous system would not be without precedent (e.g. there exist both vasoconstrictor and vasodilator sympathetic neurons, and both, inhibitor and activator enteric motoneurons). The way to this question is finally open to research, and as referee#2 says “it is early days”.

**Public review by Referee #2**

This work further documents differences between the cranial and sacral parasympathetic outflows that have been known since the time of Langley - 100 years ago.

We assume that the referee means that it is the “cranial and sacral parasympathetic outflows” which “have been known since the time of Langley”, not their differences (that we would “further document”): the differences were explicitly negated by Langley. As a matter of fact, the sacral and cranial outflows were first likened to each other by Gaskell, 140 years ago (Gaskell, 1886). This anatomic parallel (which is deeply flawed (Espinosa-Medina et al., 2018)) was inherited wholesale by Langley, who added one physiological argument (Langley and Anderson, 1895) (which has been contested many times (Espinosa-Medina et al., 2018) and references within).

In addition, the sphenopalatine and other cranial ganglia develop from placodes and the neural crest, while sympathetic and sacral ganglia develop from the neural crest alone.

Contrary to what the referee says, the sphenopalatine has no placodal contribution. There is no placodal contribution to any autonomic ganglion, sympathetic or parasympathetic (except an isolated claim concerning the ciliary ganglion (Lee et al., 2003)). All autonomic ganglia derive from the neural crest as determined a long time ago in chicken. For the sphenopalatine in mouse, see our own work (Espinosa-Medina et al., 2016).

One feature that seems to set the pelvic ganglion apart is […] the convergence of preganglionic sympathetic and parasympathetic synapses on individual ganglion cells (Figure 3). This unusual organization has been reported before using microelectrode recordings (see Crowcroft and Szurszewski, J Physiol (1971) and Janig and McLachlan, Physiol Rev (1987)). Anatomical evidence of convergence in the pelvic ganglion has been reported by Keast, Neuroscience (1995).

Contrary to what the referee says, we do not provide in Figure 3 any evidence for anatomic convergence, i.e. for individual pelvic ganglion cells receiving dual lumbar and sacral inputs. We simply show that cholinergic neurons figure prominently among targets of the lumbar pathway. This said, the convergence of both pathways on the same pelvic neurons, described in the references cited by the referee, is another major problem in the theory of the “sacral parasympathetic” (as we discussed previously (Espinosa-Medina et al., 2018)).

It should also be noted that the anatomy of the pelvic ganglion in male rodents is unique. Unlike other species where the ganglion forms a distributed plexus of mini-ganglia, in male rodents the ganglion coalesces into one structure that is easier to find and study. Interestingly the image in Figure 3A appears to show a clustering of Chat-positive and Th-positive neurons. Does this result from the developmental fusion of mini ganglia having distinct sympathetic and parasympathetic origins?

The clustering of Chat-positive and Th-positive cells could arise from a number of developmental mechanisms, that we have no idea of at the moment. This has no bearing on sympathetic and parasympathetic.

In addition, Brunet et al dismiss the cholinergic and noradrenergic phenotypes as a basis for defining parasympathetic and parasympathetic neurons. However, see the bottom of Figure S4 and further counterarguments in Horn (Clin Auton Res (2018)).

The bottom of Figure S4 simply indicates which cells are cholinergic and adrenergic. We have already expounded many times that noradrenergic and cholinergic do not coincide with sympathetic and parasympathetic. Henry Dale (Nobel Prize 1936) demonstrated this. Langley himself devoted several pages of his final treatise to this exception to his “Theory on the relation of drugs to nerve system” (Langley, 1921) (p43) (which was actually a bigger problem for him than it is for us, for reason which are too long to recount here; it is as if the theoretical difficulties experienced by Langley had been internalized to this day in the form of a dismissal of the cholinergic sympathetic neurons as a slightly scandalous but altogether forgettable oddity). (Horn, 2018) reviews the evidence that the thoracic cholinergic sympathetic phenotype is brought about by a secondary switch upon interaction with the target and argues that this would be a fundamental difference with the sacral “parasympathetic”. But in fact the secondary switch is preceded by co-expression of ChAT and VAChT with Th in most sympathetic neurons (reviewed in (Ernsberger and Rohrer, 2018)); and we have no idea of the dynamic in the pelvic ganglion. It may also be mentioned in this context that target-dependent specification of neuronal identity has also been demonstrated of other types of sympathetic neurons (Furlan et al., 2016)

What then about neuropeptides, whose expression pattern is incompatible with the revised nomenclature proposed by Brunet et al.?

There was never any neuropeptide-inspired criterion for a nomenclature of the autonomic nervous system.

Figure 1B indicates that VIP is expressed by sacral and cranial ganglion cells, but not thoracolumbar ganglion cells.

Contrary to what the referee says, there are VIP-positive cells in our sympathetic data set and even strongly positive ones, except they are scattered and few (red bars on the UMAP). They correspond to cholinergic sympathetics, likely sudomotor, which are known to contain VIP (e.g.(Anderson et al., 2006)(Stanke et al., 2006)). In other words, VIP is probably part of what we call the cholinergic synexpression group (but was not placed in it by our calculations, probably because VIP is also expressed in sympathetic noradrenergic cells, albeit at lower levels).

The authors do not mention neuropeptide Y (NPY). The immunocytochemistry literature indicates that NPY is expressed by a large subpopulation of sympathetic neurons but never by sacral or cranial parasympathetic neurons.

Contrary to what the referee says, Keast (Keast, 1995) finds 3.7% of pelvic neurons double stained for NPY and VIP in male rats, and says (Keast, 2006) that in females “co-expression of NPY and VIP is common” ( thus in cholinergic neurons that the referee calls “parasympathetic”). Single cell transcriptomics is probably more sensitive than immunochemistry, and in our dichotomized data set (table S1), NPY is expressed in all pelvic clusters and all sympathetic ganglia. In other words, it is one more argument for their kinship. It does not appear in the heatmap because it ranks below the 100 top genes.

**Answer to the original recommendations by Referee #2**

Introduction - the use of the words 'consensual' and 'promiscuity' are not clear and rather loaded in the context of the pelvic ganglia. Pick alternative words.

There is no sexual innuendo inherent in “promiscuity”: “condition of elements of different kinds grouped or massed together without order” (Oxford English Dictionary). We replaced “never consensual” by “never generally accepted”.

Results - Page 2 - what sex were the mice? Previous works indicate significant sexual dimorphism in the pelvic ganglion.

The mice included both males and females, and male and female cells are represented in all ganglia and clusters. This is now mentioned in the Material and Methods. Thus, however unsuited to analyze sexual dimorphism, our data set ensures that all the cell types we describe are qualitatively present in both sexes.

Results line 3 - the celiac and mesenteric ganglia are prevertebral ganglia and not part of the sympathetic chain. The chain refers to the paravertebral ganglia.

We replaced “part of the prevertebral chain” by “belonging to prevertebral ganglia”. This said, there are precedents for “prevertebral chain ganglia” to designate the rostro-caudal series of prevertebral ganglia. Rita Levi-Montalcini, for example, writes in 1972 “The nerve cell population of para- and prevertebral chain ganglia is reduced to 3–5% of that of controls”. (10.1016/0006-8993(72)90405-2).

Page 3 - "as the current dogma implies". Dogma often refers to opinion or church doctrine. The current nomenclature is neither. Pick another word.

There is little in science that is proven to the point of eliminating any element of opinion. “Dogma” refers to “that which is held as a principle or tenet […], especially a tenet authoritatively laid down by […] a school of thought” (OED). And “dogma” is used in science to designate tenets better experimentally supported than the “sacral parasympathetic”, such as the “central dogma of molecular biology”.

Page 3 - "To give justice" implies the classical notion is unjust. How about, 'to further explore previous evidence indicating that ....'

The term is indeed not proper English for the meaning intended, and the right expression is “to do justice”, to mean: “to treat [a subject or thing] in a manner showing due appreciation, to deal with [it] as is right or fitting” (OED). We have corrected the paper accordingly.

Page 4 top - the convergence indicated by Figure 3 does not justify excluding cholinergic and noradrenergic genes from the analysis.

Contrary to what the referee says, Figure 3 does not show any “convergence”, see our answer to Referee#1. What Figure 3 shows is that cells that are targeted by the lumbar pathway (a pathway universally deemed “sympathetic”) are cholinergic in massive proportion. Therefore, by an uncontroversial criterion, the pelvic ganglion contains lots of sympathetic cholinergic neurons. The only other option is to declare that sympathetic preganglionic neurons synapse onto parasympathetic postganglionic ones (which is what Referee#1 proposes, and considers “much simpler”. We beg to differ).

Our justification for excluding cholinergic and noradrenergic genes from the definition of “sympathetic” and “parasympathetic” is simply that sympathetic neurons can be cholinergic (to sweat glands and periosteum; and — as we show in Figure 3 — many targets of the lumbar pathway); One can also note that anywhere else in the nervous system, classifying cell types as a function of neurotransmitter phenotype would lead to non-sensical descriptions, such as putting together pyramidal cells and cerebellar granules, or motor neurons and basal forebrain cholinergic neurons. Indeed Referee#1 proposes such a revolutionary revision, by calling all cholinergic autonomic neurons “parasympathetic” (see our answer above).

Keast (1995) did similar experiments and used presynaptic lesions to draw a different conclusion indicating preferential innervation pelvic subpopulations.

Keast found “preferential” innervation of pelvic subpopulations based on lesion experiments; Nevertheless, she concluded (at the time) that “the correct definition of these two components of the nervous system is based on neuroanatomy rather than chemistry” (Keast, 2006).

Page 4 - "In the aggregate, the pelvic ganglion is best described as a divergent sympathetic ganglion devoid of parasympathetic neurons" The notion of a divergent ganglion is completely unclear!

We take “divergent” in a developmental or evolutionary meaning: related to sympathetic ganglia, yet somewhat differing from them. Elsewhere we use the word “modified”. Importantly (and as cited in the paper), a similar situation emerges from the single cell transcriptomic analysis of the lumbar and sacral preganglionics (by other research groups).

Granted, it is devoid of neurons having the signature of cranial parasympathetics, but that is insufficient to conclude that they are not parasympathetics.

If a genetic signature which is not only un-parasympathetic, but sympathetic-like remains compatible with some version of the label “parasympathetic”, we get dangerously close to dismissing the molecular make-up of a neuron as a definition of its type. This goes against any contemporary understanding of neuron types (take (Zeisel et al., 2018) among hundreds of other examples).

Page 4 - "the entire taxonomy of autonomic ganglia could be a developmental readout of Hox genes." This reader completely agrees! We appreciate this would be difficult to test but it helps to explain possible differences along the rostro-caudal axis. Consider making this a key implication of the study!

If the reader agrees, then his/her previous points become mysterious: we speculate that the Hox code determines the structure of the autonomic nervous system, i.e. the array, along the rostrocaudal axis, of a bulbar parasympathetic, a thoracolumbar sympathetic and lumbo-sacral “pelvo-sympathetic”. The existence of caudal parasympathetic neurons, on the contrary, would subvert any role for Hox genes: similar neurons (similar enough to be called by the same name) would arise at completely different rostro-caudal levels, i.e. with a different Hox code.

Page 5 - "It is thus remarkable ...that we uncover in no way contradicts the physiology." Not really. The 'classical' sympathetic system innervates the limbs, and the skin and it participates in thermoregulation and in cardiovascular adjustments to exercise. The parasympathetic system does none of these things. Reclassing the pelvic outflow as pseudo-sympathetic contradicts this physiology.

We do not say that the sacral outflow is classically sympathetic; We go all the way to proposing the special name “pelvo-sympathetic”; And we insist that these special sympathetic-like neurons have special targets (detrusor muscle, helicine arteries…): there is no contradiction. Not only is there no contradiction, but we remove the mind-twister of an anatomical/genetic/cell type-based “sacral parasympathetic” combined with a lack of physiological lumbosacral antagonism (we provide a short history of this dissonance in (Espinosa-Medina et al., 2018)), which led Wilfrid Jänig to write (Jänig, 2006)(p. 357): “Thus, functions assumed to be primarily associated with sacral (parasympathetic) are well duplicated by thoracolumbar (sympathetic) pathways. This shows that the division of the spinal autonomic systems into sympathetic and parasympathetic with respect to sexual functions is questionable”. We could not agree more: this division is questionable in terms of physiology and inexistent in terms of cell types. In other words, we reconcile cell types with physiology (but “it is early days”).

**Answer to the novel recommendations by Referee #2**

In addition to my original comments, important anatomical and functional distinctions are not explained by the data in this paper. ANATOMY- Sympathetic ganglia are located in close proximity to major branches of the aorta. Cranial and sacral parasympathetic ganglia are located next to or within the structures they innervate (e.g. eye, lung, heart, bladder).

The pelvic ganglion, including some of its cholinergic neurons, that the referee insist are parasympathetic, is further removed from one of its major targets (the helicine arteries of the external genitals) than the sympathetic prevertebral ganglia are of some of theirs (like the gut or kidney). We discussed this issue in (Espinosa-Medina et al., 2018).

FUNCTION- The sympathetic system controls state variables (e.g. body temperature, blood pressure, serum electrolytes and fluid balance), parasympathetic neurons do not.

Even in the classical view, the sympathetic system controls the blood vessels of the external genitals or the size of the pupil, for example, which are not state variables.

[…] The data in the paper are a useful next step in defining the genetic diversity of autonomic neurons but do not justify or improve upon existing nomenclature. The future challenge is to understand distinctions between subsets of autonomic ganglion cells that innervate different targets and the principles that govern the integrative function of the autonomic motor system that controls behavior.

We thank the referee for finding our data useful; and we fully agree with the latter statement. However, neurons, like many other cell types, are hierarchically organized (Zeng and Sanes, 2017), i.e. subsets of neurons belong to sets, with defining traits. Our data argue that there is no parasympathetic neuronal set that includes any pelvic ganglionic neuron. In contrast, there is a ganglionic sympathetic set (defined by our analysis of gene expression) which includes all of them — as there is a preganglionic sympathetic set that includes sacral preganglionics (Alkaslasi et al., 2021; Blum et al., 2021)(although the direct comparison with cranial preganglionics is yet to be made).

**References**

Anderson, C. R., Bergner, A. and Murphy, S. M. (2006). How many types of cholinergic sympathetic neuron are there in the rat stellate ganglion? Neuroscience 140, 567–576.

Alkaslasi, M. R., Piccus, Z. E., Hareendran, S., Silberberg, H., Chen, L., Zhang, Y., Petros, T. J. and Le Pichon, C. E. (2021). Single nucleus RNA-sequencing defines unexpected diversity of cholinergic neuron types in the adult mouse spinal cord. Nat Commun 12, 2471.

Blum, J. A., Klemm, S., Shadrach, J. L., Guttenplan, K. A., Nakayama, L., Kathiria, A., Hoang, P. T., Gautier, O., Kaltschmidt, J. A., Greenleaf, W. J., et al. (2021). Single-cell transcriptomic analysis of the adult mouse spinal cord reveals molecular diversity of autonomic and skeletal motor neurons. Nat Neurosci 24, 572–583.

Ernsberger, U. and Rohrer, H. (2018). Sympathetic tales: subdivisons of the autonomic nervous system and the impact of developmental studies. Neural Dev 13, 20.

Espinosa-Medina I, Saha O, Boismoreau F, Chettouh Z, Rossi F, Richardson WD, Brunet JF (2016) The sacral autonomic outflow is sympathetic. Science 354, 893-897

Espinosa-Medina, I., Saha, O., Boismoreau, F. and Brunet, J.-F. (2018). The “sacral parasympathetic”: ontogeny and anatomy of a myth. Clin Auton Res 28, 13–21.

Furlan, A., La Manno, G., Lübke, M., Häring, M., Abdo, H., Hochgerner, H., Kupari, J., Usoskin, D., Airaksinen, M. S., Oliver, G., et al. (2016). Visceral motor neuron diversity delineates a cellular basis for nipple- and pilo-erection muscle control. 19, 1331–1340.

Gaskell, W. H. (1886). On the Structure, Distribution and Function of the Nerves which innervate the Visceral and Vascular Systems. J Physiol 7, 1-80.9.

Horn, J. P. (2018). The sacral autonomic outflow is parasympathetic: Langley got it right. Clin Auton Res 28, 181–185.

Jänig, W. (2006). The Integrative Action of the Autonomic Nervous System: Neurobiology of Homeostasis. Cambridge: Cambridge University Press.

Keast, J. R. (1995). Visualization and immunohistochemical characterization of sympathetic and parasympathetic neurons in the male rat major pelvic ganglion. Neuroscience 66, 655–662.

Keast, J. R. (2006). Plasticity of pelvic autonomic ganglia and urogenital innervation. International Review of Cytology - a Survey of Cell Biology, Vol 248 248, 141-+.

Langley, J. N. (1921). In The autonomic nervous system (Pt. I)., p. Cambridge: Heffer & Sons ltd.

Langley, J. N. and Anderson, H. K. (1895). The Innervation of the Pelvic and adjoining Viscera: Part II. The Bladder. Part III. The External Generative Organs. Part IV. The Internal Generative Organs. Part V. Position of the Nerve Cells on the Course of the Efferent Nerve Fibres. J Physiol 19, 71–139.

Lee, V. M., Sechrist, J. W., Luetolf, S. and Bronner-Fraser, M. (2003). Both neural crest and placode contribute to the ciliary ganglion and oculomotor nerve. Developmental biology 263, 176–190.

Stanke, M., Duong, C. V., Pape, M., Geissen, M., Burbach, G., Deller, T., Gascan, H., Parlato, R., Schütz, G. and Rohrer, H. (2006). Target-dependent specification of the neurotransmitter phenotype:cholinergic differentiation of sympathetic neurons is mediated in vivo by gp130 signaling. Development 133, 141–150.

Zeisel, A., Hochgerner, H., Lönnerberg, P., Johnsson, A., Memic, F., van der Zwan, J., Häring, M., Braun, E., Borm, L. E., La Manno, G., et al. (2018). Molecular Architecture of the Mouse Nervous System. Cell 174, 999-1014.e22.

Zeng, H. and Sanes, J. R. (2017). Neuronal cell-type classification: challenges, opportunities and the path forward. Nat Rev Neurosci 18, 530–546.